# Marine Biofilm Model Comprising a Loop-Type Biofilm Reactor and a *Halomonas* Strain HIG FST4 1, an Active Biofilm-Forming Bacterium

**Akiko Ogawa** [1,*] , **Shoya Hosaka** [2], **Hideyuki Kanematsu** [3] **and Michiko Yoshitake** [4]

1   Department of Chemistry and Biochemistry, National Institute of Technology (KOSEN), Suzuka College, Suzuka 510-0294, Japan
2   Advanced Engineering Course of Science and Technology for Innovation, National Institute of Technology (KOSEN), Suzuka College, Suzuka 510-0294, Japan
3   Joint Research Center between Academia and Industries, National Institute of Technology (KOSEN), Suzuka College, Suzuka 510-0294, Japan
4   National Institute for Materials Science (NIMS), Tsukuba 305-0044, Japan
*   Correspondence: ogawa@chem.suzuka-ct.ac.jp; Tel.: +81-5-9368-1768

**Abstract:** In ocean and coastal waters, the formation of biofilms on artificial matters often causes intractable phenomena such as the deterioration of surface functions and corrosion, resulting in significant economic damage. Thus, methods for inhibiting biofilm formation are in high demand, and many new anti-biofilm products are being designed on a daily basis. However, practical and safe assays for evaluating anti-biofilm formation have not yet been established. In this study, we developed a more practical and safer biofilm formation test system composed of a loop-type laboratory biofilm reactor (LBR) and HIG FST4 1, a *Halomonas* strain derived from ballast seawater, in comparison with a slowly rotating test-tube culture (TTC) test. To evaluate biofilm formation in an LBR and TTC, three materials (pure iron, pure aluminum, and soda lime glass) were tested, and Raman spectroscopic analysis was used for the identification and relative quantification of the biofilm contents. Regardless of the test method, biofilm formation progressed in the order of soda lime glass < pure aluminum < pure iron. The Raman peaks showed that the LBR test samples tended to remove polysaccharides compared to the TTC test samples and that the proportion of proteins and lipids in the LBR test samples was much higher than that of the TTC test samples. These results show that the combination of HIG FST4 1 and LBR is suitable for biofilm formation in a practical marine environment.

**Keywords:** biofilm model; biofilm formation; marine environment; *Halomonas*; laboratory-biofilm reactor

## 1. Introduction

In marine environments, biofilm formation contributes to the biofouling and biocorrosion of the bottom of marine vessels, cooling pipes in power plants, and offshore oilfield structures, leading to a large loss of energy and the deterioration of marine structures and marine vessels [1–6]. For the protection of materials from such economically significant problems, the prevention of biofilm formation has received much attention. Several promising methods have been proposed for regulating the growth of biofilms on materials such as steel, brass, and cement, including using anti-fouling paints incorporated with natural compounds isolated from seaweeds, marine invertebrates, and marine organisms [2], using silicone/zinc oxides nanorod composite coatings [7] and using copper- and silver-based coatings [8,9].

A practical marine test is considered to be the best way to evaluate biofilm formation quantitatively and qualitatively as it provides insights into the extent of biofilm formation inhibition on treated materials in the marine environment. However, the results of a practical marine test may be influenced by environmental factors, such as weather conditions,

water temperature, and geography, resulting in the repetition of tests and delays in research. Therefore, a stable and simple laboratory test is needed for the controlled evaluation of the anti-biofilm activity of materials.

*Pseudomonas aeruginosa* and *Staphylococcus aureus* are often used as model bacteria to study biofilm formation in laboratories as they show an enhanced biofilm forming ability. However, these bacteria exhibit several limitations. They are major pathogens of chronic infectious diseases, which makes their handling too dangerous for researchers, including inexperienced students. Moreover, these bacteria are usually cultured in conditions mimicking human body conditions, which are remarkably different from marine conditions. On the other hand, *Halomonas* is a halophilic bacterium that is relatively safe and is attracting much attention from the viewpoint of industrial use; therefore, we aimed to evaluate whether *Halomonas* can be used as a model organism for studying biofilm formation instead of *P. aeruginosa* and *S. aureus*.

Furthermore, in this study, we aimed to develop a reliable and safe laboratory marine biofilm model comprising a loop-type laboratory biofilm reactor (LBR) and HIG FST4 1, an *Halomonas* strain isolated from a ballast fuel tank at the University of Oklahoma [10]. Simultaneously, we aimed to perform a slowly rotating test-tube culture (TTC) to produce biofilms and compare their features to those of biofilms formed in the LBR. Pure iron, pure aluminum, and soda lime glass were used as a corrosive material, a marine architectural material, and an anti-biofilm material, respectively. Moreover, we attempted to conduct a Raman spectroscopic analysis to confirm the formation of biofilms on the sample materials.

## 2. Materials and Methods

### 2.1. Materials

A pure iron (Fe) plate (0.5 cm thick; Nilaco, Tokyo, Japan), an aluminum (Al) plate (0.5 cm thick; Nilaco), and a soda lime glass slide (1 mm thick; As One, Osaka, Japan) were cut into 1 cm long and 2 cm wide pieces. The surface of each piece was polished using a 1000-grit emery paper, followed by washing with 99.5% ethanol, and immersion in a 70% ethanol solution. Prior to the biofilm-formation experiments, the materials were air-dried on a clean bench to avoid interference from the ethanol solution.

### 2.2. Bacterium and Its Preculture

HIG FST4 1 was streaked on Marine Broth 2216 agar plates (Thermo Fisher Scientific, Waltham, MA, USA) for 48 h at 25 °C. Then, one colony of HIG FST4 1 was transferred to a test tube containing 5 mL of Marine Broth 2216 (Thermo Fisher Scientific) and cultured for 24 h at 25 °C. Next, 500 μL of the culture solution was transferred to a test tube containing 5 mL of Marine Broth 2216 and incubated for 4 h at 25 °C. Next, 200 μL of the culture solution was transferred to 400 mL of four-fold diluted Marine Broth 2216, which was used for both the rotating and circulatory biofilm formation tests.

### 2.3. Biofilm Formation Tests

HIG FST4 1 culture that was incubated for 4 h (200 μL) was added to 400 mL of four-fold diluted Marine Broth 2216. This was referred to as the starter culture. A 15 mL centrifuge tube (As One) was filled with 4 mL of the starter culture and one piece of material was added. The centrifuge tubes were rotated at 50 rpm for two weeks at 25 °C. This was designated as the slowly rotating TTC test. The remaining starter culture (396 mL) was transferred to a sterile three-necked beaker that was a part of a closed-circuit-type biofilm reactor (Figure 1). The starter culture was circulated through the sterile tubes of the reactor using a tubing pump (As One) at a rate of 5 mL/s (Reynolds number: Re = 396) at 25 °C for 3 days. This was designated as the LBR test. After the completion of the slowly rotating TTC and LBR tests, each piece of material was taken out and soaked in a 30% ethanol–water solution. Every 15 min, the soaking solution was gradually changed to a higher concentration ethanol–water solution (50%, 60%, 70%, 80%, 90%, 95%, and 99.5%). Next, the ethanol solution was exchanged with an ethanol/t-butanol solution (7/3,

5/5, 3/7, and 0/100). Subsequently, the frozen pieces of the materials were dried using a reduced pressure drying apparatus (As One).

(a)

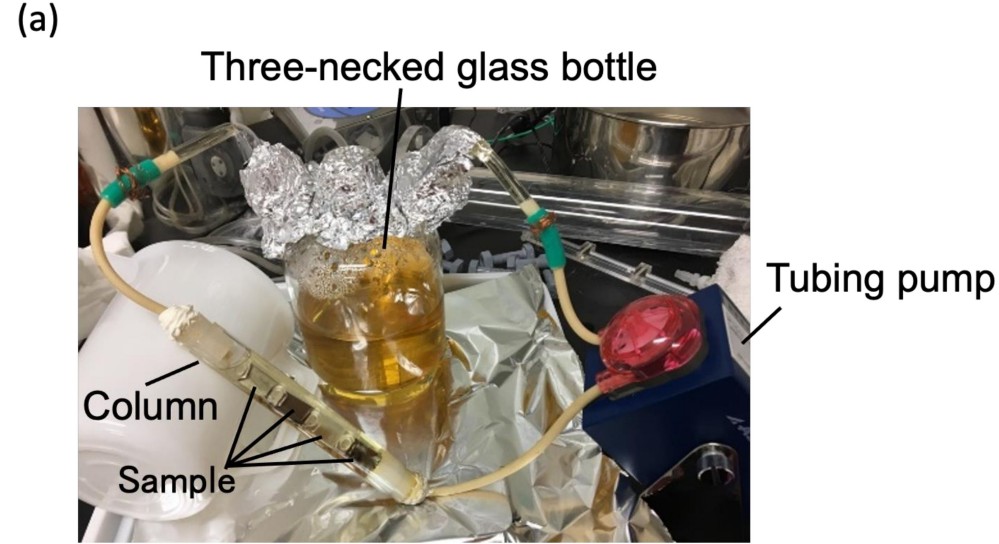

(b)

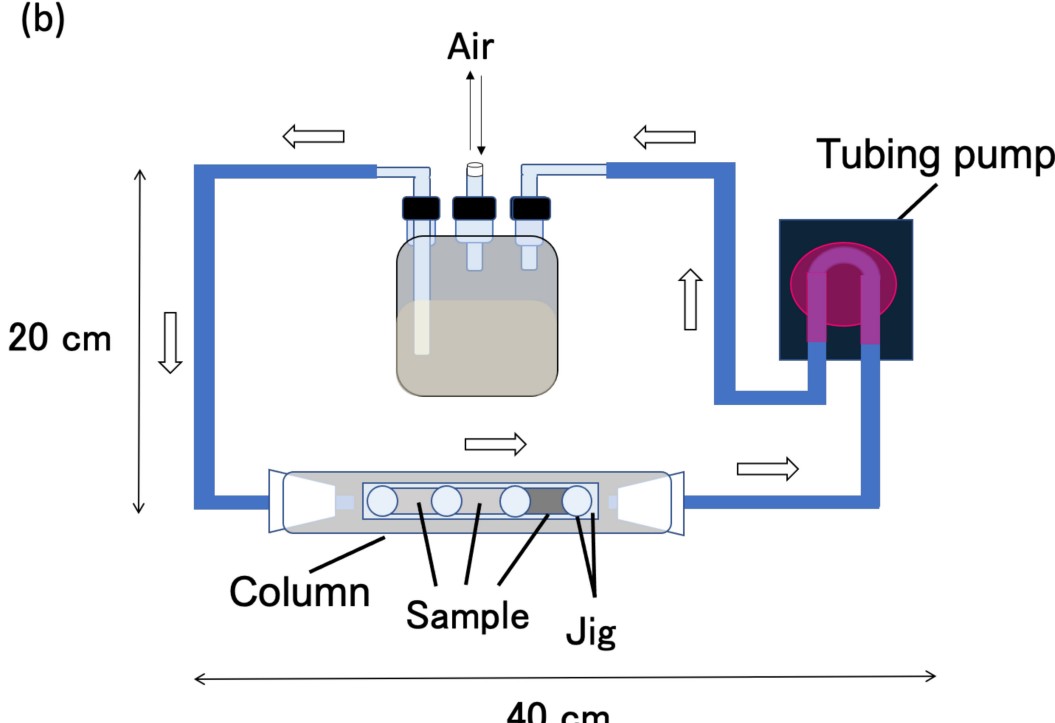

**Figure 1.** Outline of the closed-circuit biofilm reactor. (**a**) and (**b**) indicate the photograph of practical reactor and the scheme of this reactor, respectively. In the illustration (**b**), arrows indicate the flow direction. Two necks of the three-necked beaker are connected to glass tubes, and the last (center position) is used for aseptic air exchange.

*2.4. Raman Spectroscopy for Confirming Biofilm Formation*

Biofilms are composed of bacterial cells and secreted extracellular polymeric substrates (EPS) such as proteins, nucleic acids, polysaccharides, and lipids. These compounds can be detected using a Raman spectrometer that detects specific covalent bonds, such as amide bonds in proteins, carbonyl bonds in proteins and lipids, and hydroxyl bonds in polysaccharides and nucleic acids. First, each piece of material was observed under a

100-fold condition using the accessory bright-field microscope of a Raman spectrometer (NRS-3100, JASCO, Tokyo, Japan) to detect debris on the surface of the materials. When debris was detected, the spot was exposed to a 532.10 nm laser (1.7 mW) ten times for 3 s and was scanned from 600 to 3800 cm$^{-1}$. The obtained Raman spectra were used to identify the type of chemical bonds as previously described in [11–17]. Handling was randomly performed at five positions on the surface of each piece of material.

## 3. Results and Discussion

Practically, steels and aluminum alloys are used for marine constructs and vessels; using these alloys ran the risk that other compounds would affect the biofilm formation. Therefore, we selected three materials: pure Fe as a corrosive and easy biofilm-forming substrate in marine environments, soda lime glass as an anti-accumulating substrate for biofilms, and pure Al as a prospective marine artificial product. Fe is an essential element for most organisms. Fe is reported to accelerate biofilm formation by some bacteria [18–20] and is the key factor for phytoplankton growth in marine environments [21]. The surface of soda lime glass is usually smooth and is an important factor that aids the easy decoupling of debris and biofilms from surfaces by flowing water. Therefore, soda lime glass has been used to protect materials against dust. Al is easily oxidized in an aqueous environment, producing an oxidative coating that protects the inner part of the metal. The oxidative coated Al is suggested to protect against corrosion. Additionally, Al and its alloys are lighter than major steel alloys used in ships, bridges, and other marine constructions, reducing the running cost owing to its weight. Therefore, Al, instead of steel alloys, is a prospective material for marine environments.

### 3.1. Differences in Surface of Materials before and after Slowly Rotating TTC and LBR Tests

Figure 2 shows the surface of materials before and after the slowly rotating TTC and LBR tests. Fe and Al displayed several unidirectional lines as a result of a prior polishing using emery paper.

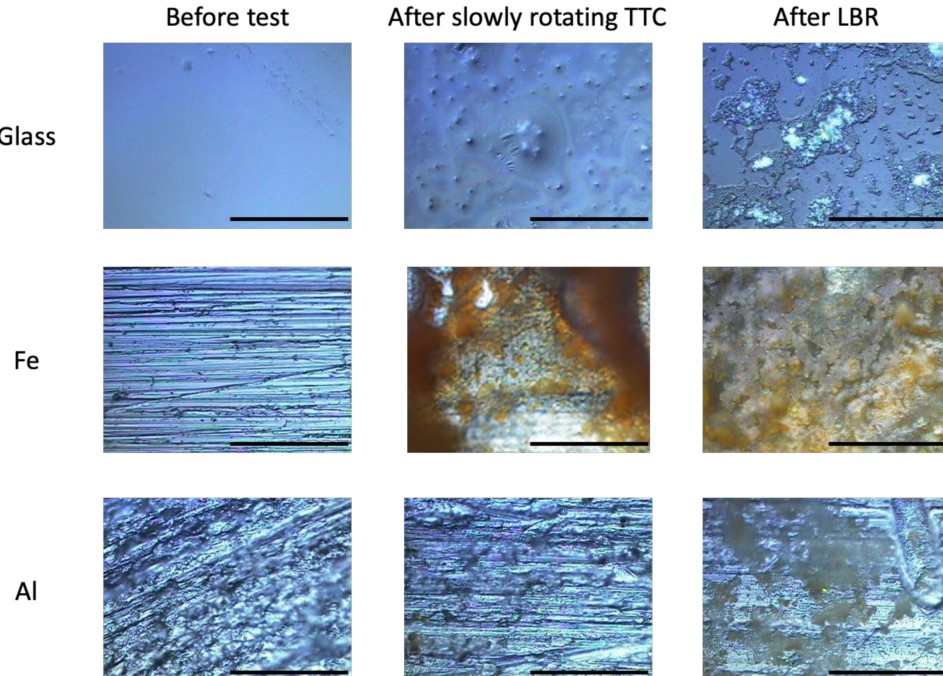

**Figure 2.** Optical microscopic images of the surface of materials that were analyzed by Raman spectroscopy. These samples were observed under an optical microscope equipped with a Raman spectroscopy apparatus ($\times$100). The center of each image indicates the position analyzed by Raman spectroscopy. Al: aluminum; Fe: iron; LBR: laboratory biofilm reactor; and TTC: test-tube culture. Each black bar indicates 20 μm.

Prior to the tests, the surface of the soda lime glass was smooth, the surface of the Fe had small pits resembling corroded sites, and the surface of the Al appeared rough due to the oxidized and non-oxidized aluminum parts. After the slowly rotating TTC and LBR tests, islet-like sediments were observed on the surface of the soda lime glass pieces. Brown-colored slimy sediments covered the surface of the Fe pieces. Yellow-brown colored matter was observed to partially accumulate on the surface of the Al pieces. Collectively, these results show that corroded matter and culture-derived matter, including *Halomonas*, EPS, and its metabolites, were accumulated on the surface of soda lime glass, Fe, and Al.

### 3.2. Analysis of Raman Spectra

Biofilms are composed of bacterial cells and EPS, such as polysaccharides, proteins, nucleic acids, and lipids [22]. The Raman peaks of biofilms are usually multiple and complicated, and major matters tend to be detected strongly as 'major peaks', thereby enabling the identification of the main contents of the biofilm formed on each piece of material.

### 3.2.1. Pure Iron Samples

Figure 3 shows the Raman spectra of the surface of Fe pieces before and after the slowly rotating TTC (Fe-TTC) and LBR (Fe-LBR) tests. No large peaks were observed for Fe prior to the tests. The large peak at 746 cm$^{-1}$ (Fe-TTC) and at 743 cm$^{-1}$ (Fe-LBR) was associated with $\gamma$-FeO(OH), which is known as a rust in water. $\gamma$-FeO(OH) exists easily as high chloride ions [23], which are abundant in Marine Broth 2216 (the culture medium). Considering that the color of $\gamma$-FeO(OH) is yellowish-brown [24], Fe-TTC and Fe-LBR were corroded. Some major peaks were detected in both Fe-TTC and Fe-LBR at 1000–1700 cm$^{-1}$. For Fe-LBR, the peak at 1301 cm$^{-1}$ was the largest and was attributed to the HC=CH bond of unsaturated fatty acids [16]. Furthermore, a small peak at 910 cm$^{-1}$ was attributed to the stretched P–O bond in nucleic acids. The peaks at 1064 cm$^{-1}$ and 1147 cm$^{-1}$ were related to the scissoring and twisting vibrations of the methyl groups in the lipids [16]. The broad peak at 1400–1478 cm$^{-1}$ was attributed to the scissoring and twisting vibrations of CH$_2$ in the lipids. The peak at 1529 cm$^{-1}$ corresponds to the vibration of the peptide bond (stretched C=O bond) in the proteins [16]. The peak at 1593 cm$^{-1}$ was attributed to humic-like substances [12]. For Fe-TTC, the largest peak, which was at 1593 cm$^{-1}$, and the second largest peak, which was at 1340 cm$^{-1}$, were associated with humic-like substances [12]. The small peak at 850 cm$^{-1}$ was attributed to the anomeric skeletal configuration of polysaccharides [11]. The peaks at 1901 cm$^{-1}$, 1190 cm$^{-1}$, and 1417–1438 cm$^{-1}$ corresponded to the vibration of a stretched C–C bond in the fatty acids, C–O–O stretching vibrations, and the scissoring and twisting vibrations of CH$_2$, respectively. These peaks were attributed to the lipids [13,17]. The 1516 cm$^{-1}$ peak was attributed to amide linkage I in the protein. Over the 1700 cm$^{-1}$ region, two broad middle peaks (2866–2905 cm$^{-1}$ and 3254–3538 cm$^{-1}$) were observed for Fe-LBR. They provide fundamental vibrational information on proteins and nucleic acids [17]. In contrast, a very wide middle peak was observed from 1769 cm$^{-1}$ to 3665 cm$^{-1}$ for Fe-TTC. In particular, the 2871–2952 cm$^{-1}$ peak was relatively sharpened, which was related to the fundamental vibration of proteins and nucleic acids. No polysaccharide-related peaks were detected for Fe-LBR. Collectively, these results suggested that biofilm was formed on the surfaces of both Fe-TTC and Fe-LBR and that the biofilms were rich in Fe rust. However, the major components of these biofilms were different between Fe-TTC and Fe-LBR. In Fe-TTC, humic-like acids were the main component, proteins and nucleic acids were relatively abundant, lipids were present, and polysaccharides were small. On the other hand, in Fe-LBR, lipids were the main matter, proteins and nucleic acids were relatively abundant, and humic-like acids were modestly present.

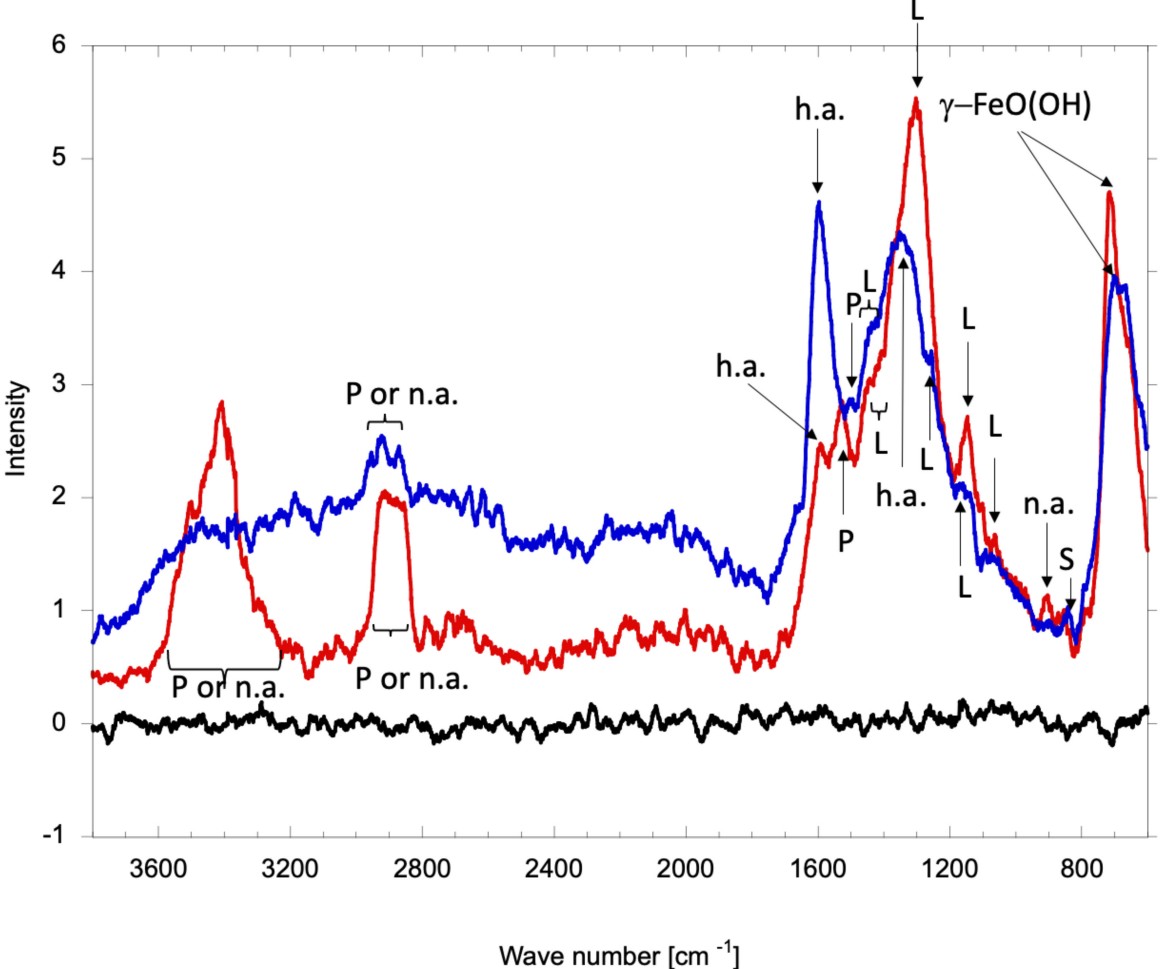

**Figure 3.** Raman spectra of the surface of pure iron pieces. The black, red, and blue lines indicate the Raman spectra for iron piece prior to the tests, after LBR test, and after slowly rotating TTC test, respectively. P: proteins; n.a.: nucleic acids; h.a.: humic-like acids; L: lipids; and S: polysaccharides.

Humic acids are macromolecular decayed organic matter distributed in terrestrial soil, natural water, and sediments [25]. Based on the characteristics of humic acids, some polymers, including polysaccharides, lipids, and proteins, are degraded by bacteria, resulting in the formation of humic acids. In this study, HIG FST4 1, a *Halomonas* sp., was used for the biofilm-formation tests. This bacterium can degrade a variety of straight-chain alkanes [26], i.e., HIG FST4 1 can partially degrade polysaccharides and lipids in EPS. Therefore, it was considered that humic-like acids in the biofilms of Fe-TTC and Fe-LBR were produced by HIG FST4 1; the source of humic-like acids was the EPS of the biofilms. Additionally, the abundance ratio of humic acids reflects the progression of biofilm formation; the higher the ratio of humic acids in biofilms, the more progressive is the biofilm formation. The biofilm of Fe-TTC contained rich humic-like acids, while that of Fe-LBR contained moderate amounts of humic-like acids, indicating that biofilm formation in Fe-TTC progressed more than that in Fe-LBR.

### 3.2.2. Soda Lime Glass Samples

Figure 4 shows the Raman spectra of the surface of soda lime glass before and after the slowly rotating TTC (Glass-TTC) and LBR (Glass-LBR) tests. Prior to the test, only one peak ($1076$–$1094$ $cm^{-1}$) was observed. This peak was attributed to the Si–O or Si–O$^-$ stretching vibration [27]. Additionally, we detected the largest peaks at $1094$ $cm^{-1}$ and $1096$ $cm^{-1}$ in the Glass-TTC and Glass-LBR, respectively. The second largest peak was observed at $600$ $cm^{-1}$ in both the Glass-TTC and Glass-LBR, which was attributed to a Si–O–Si bending

vibration in the depolymerized structural units [27]. These peaks are associated with basal soda lime glass. In addition to the two large peaks, several broad and small peaks were observed at approximately 750 cm$^{-1}$ and 1200–3800 cm$^{-1}$ both in the Glass-TTC and Glass-LBR. The C–H wagging vibration related to the polysaccharides or lipids was denoted by the peaks at 739–812 cm$^{-1}$ (Glass-TTC), 724–787 cm$^{-1}$ and 846 cm$^{-1}$ (Glass-LBR). In both the Glass-TTC and Glass-LBR, several small peaks in the range of 1200–2500 cm-1 failed to reach peak detection due to a comparably high noise, which indicates that the quantity of the substances on soda lime glass is rather low. The fundamental vibrations on the proteins or nucleic acids were denoted by two peaks at 2867–2936 cm$^{-1}$ and 3044–3566 cm$^{-1}$ for the Glass-TTC, and by three peaks at 2640–2723 cm$^{-1}$, 2862–3007 cm$^{-1}$, and 3243–3565 cm$^{-1}$ for the Glass-LBR [11]. Comparing the shape of the Raman peaks of the Glass-TTC with that of the Glass-LBR, the former was more recognizable than the latter, which indicated that the components (matter) of the biofilms on the Glass-TTC were more in amount than those on the Glass-LBR. However, since the main Raman peaks were linked to the soda lime glass itself, both the Glass-TTC and Glass-LBR displayed small amounts of biofilm.

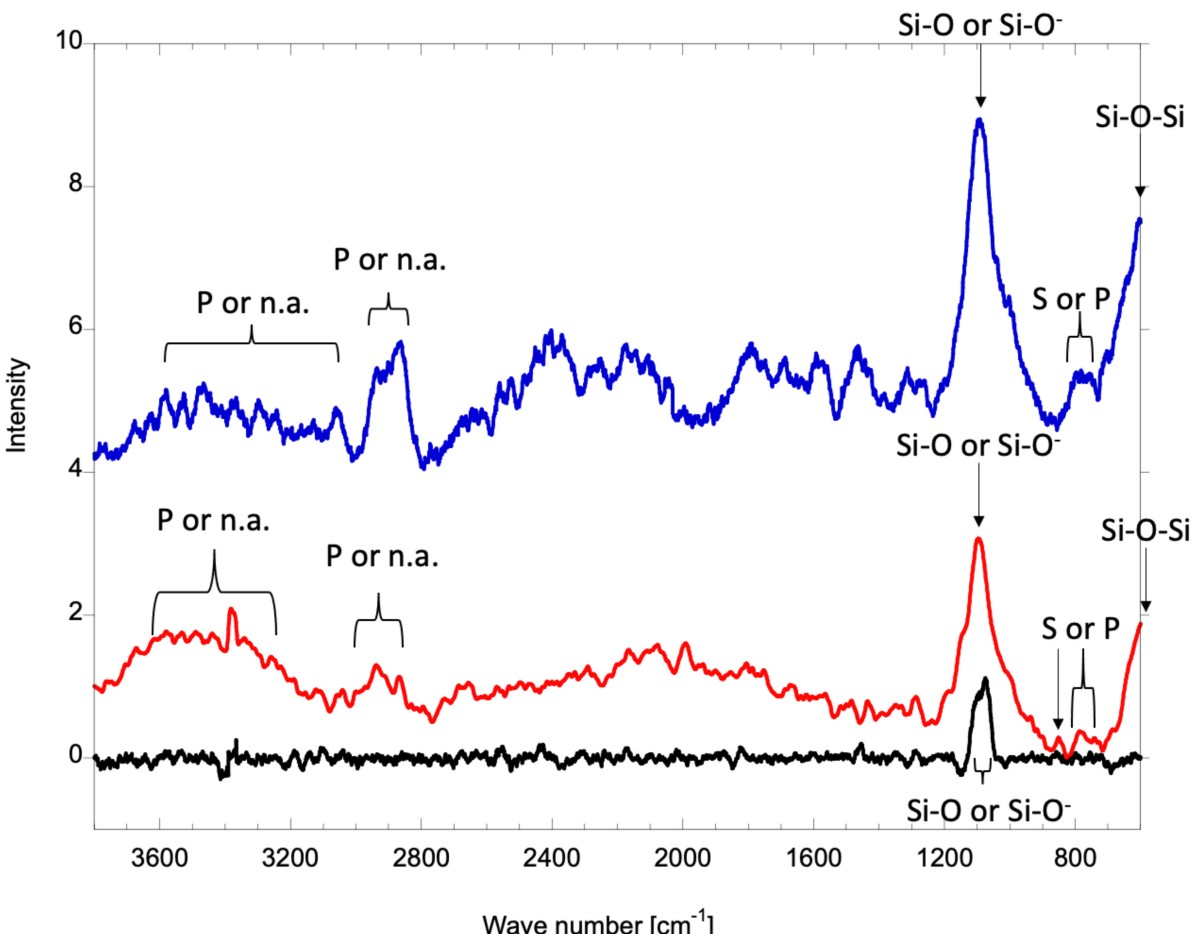

**Figure 4.** Raman spectra of the surface of soda lime glass. The black, red, and blue lines indicate the Raman spectra for iron piece prior to the tests, after LBR test, and after slowly rotating TTC test, respectively. P: proteins; n.a.: nucleic acids; h.a.: humic-like acids; L: lipids; and S: polysaccharides.

### 3.2.3. Pure Aluminum Samples

Figure 5 shows the Raman spectra of the surface Al pieces prior to the tests and after the slowly rotating TTC (Al-TTC) and LBR (Al-LBR) tests. Prior to the tests, only one peak was observed at approximately 1832 cm$^{-1}$, which was not related to the Raman peak, but was attributed to the roughness of the specimen or oxidization of the Al piece. For Al-TTC, the largest peaks at 837 cm$^{-1}$ and 1093 cm$^{-1}$ were related to the DNA back-

bone. A sharp peak at 948 cm$^{-1}$ was attributed to the anomeric skeletal configuration of the polysaccharides [11]. The peaks at 1288 cm$^{-1}$ and 1434 cm$^{-1}$ denoted the HC=CH bonds in unsaturated fatty acids. The peak at 2291–2407 cm$^{-1}$ was associated with the mixture of C=C or C=O vibrations in fatty acids, amide bonds in proteins, and DNA strand bonds [13–15]. The peaks at 2851–2924 cm$^{-1}$, 3207–3353 cm$^{-1}$, and 3450–3580 cm$^{-1}$ were attributed to the fundamental vibrations in the proteins or nucleic acids [17]. For Al-LBR, the peak at 998 cm$^{-1}$ and the largest broad peak (1193–1140 cm$^{-1}$) were attributed to the C–O–O stretching vibrations in the lipids [16]. The peaks at 1244–1324 cm$^{-1}$ corresponded to the HC=CH bonds in unsaturated fatty acids [13]. The peaks at 1508 cm$^{-1}$ and 1604–1612 cm$^{-1}$ were related to the C=O stretching vibration in the peptide bonds in the proteins [16]. The broad peak at 1784–1862 cm$^{-1}$ was attributed to carbonyl compounds in polysaccharides [11]. The peaks at 1917–1979 cm$^{-1}$ and 2102–2422 cm$^{-1}$ are associated with a mix of C=C or C=O bond vibrations in fatty acids, amide bonds in proteins, and DNA strand bonds [13–15]. The peaks at 2684–2717 cm$^{-1}$, 2883–3011 cm$^{-1}$, and 3288–3533 cm$^{-1}$ denote fundamental vibrations in proteins and nucleic acids [17]. The remarkable Raman spectra for Al-TTC and Al-LBR indicated that biofilms on Al-TTC mainly contained nucleic acids, while biofilms on Al-LBR mainly contained lipids. Moreover, it indicated that the proportion of polysaccharides in Al-TTC was higher than that in biofilms on Al-LBR.

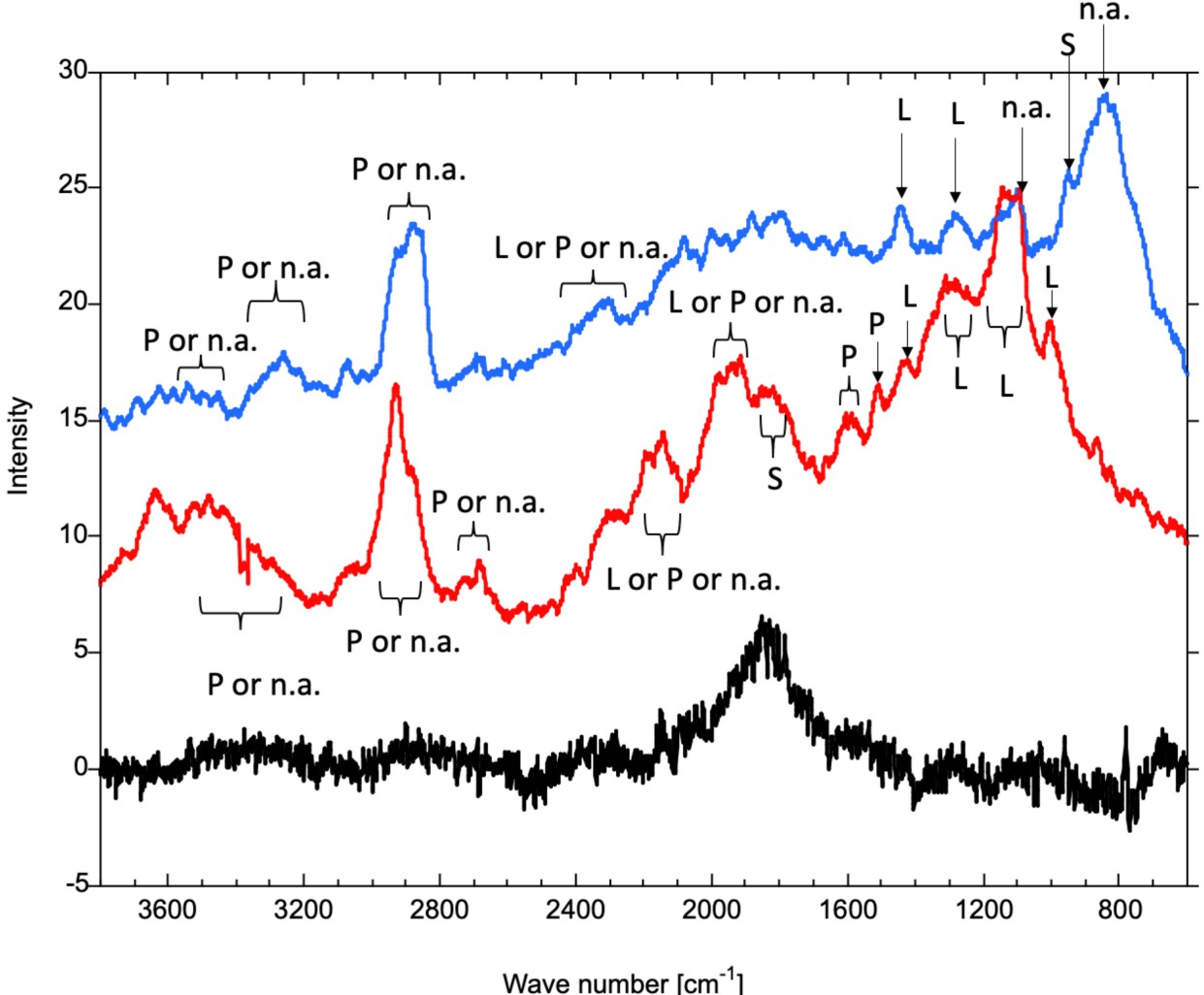

**Figure 5.** Raman spectra of the surface of pure aluminum samples. The black, red, and blue lines indicate the Raman spectra for iron piece prior to the tests, after LBR test, and after slowly rotating TTC test, respectively. P: proteins; n.a.: nucleic acids; h.a.: humic-like acids; L: lipids; and S: polysaccharides.

### 3.2.4. Comparing Progression Biofilm Formation in LBR Test with That in Slowly Rotating TTC Test

Polysaccharides act as glue, an environmental barrier, and stabilize the structure of the biofilms [28]. Polysaccharide-related Raman peaks have been reported at 580 cm$^{-1}$ [12], 850–950 cm$^{-1}$ [11], and 1550–1900 cm$^{-1}$ [11]. In this study, all materials subjected to the slowly rotating TTC test exhibited stronger polysaccharide-related Raman peaks than that which were subjected to the LBR test. This indicates that the extracellular polysaccharides were removed in a flow rate-dependent manner. On the other hand, except for the soda lime glass, the lipid-related and protein-related peaks for Fe and Al after the LBR test were relatively sharpened compared to those after the slowly rotating TTC test, which were considered to be richer in the lipids and proteins attributed to sticky biofilms.

The largest peak for glass was attributed to the soda-lime glass itself, the base material, for both the LBR and slowly rotating TTC tests. The largest peak for Fe was attributed to Fe rust in both the LBR and slowly rotating TTC tests. The largest peak for Al was attributed to the EPS of biofilms in both the LBR and slowly rotating TTC tests. Therefore, the amounts of biofilms formed on the specimens were sorted in descending order (in both slowly rotating TTC and LBR tests) as follows: glass < Fe < Al. Furthermore, Fe-TTC and Fe-LBR showed an increased corrosion and Fe rust was observed to be present in the biofilms.

In this study, HIG FST4 1 was used as a marine biofilm producer. This bacterium, which was isolated from the seawater stored in a ballast tank of an American navy vessel at the laboratory of University of Oklahoma, was grown in Marine Broth 2216, which is suitable for growing marine bacteria. This bacterium is known to form biofilms as well as decompose a variety of straight-chain alkanes [26]. Compared with *P. aeruginosa* and *S. aureus*, which are common model biofilm producers, HIG FST4 1 is not only more suitable for application in marine biofilm research, but also safer to use. The LBR test method has some advantages over the slowly rotating TTC method: (1) a shortened test period (14 days to 3 days); (2) it approaches the experimental condition more closely to a practical one; and (3) it is easy to visually manage because users can easily observe the test pieces in the column of the loop-type LBR. Therefore, the test system comprising the LBR and HIG FST4 1 will be a safe and practical model to study the biofouling of several materials such as anti-biofilm-forming materials and corrosive materials.

**Author Contributions:** Conceptualization, H.K.; methodology, S.H.; validation, A.O. and H.K.; investigation, S.H.; data curation, S.H. and H.K.; writing—original draft preparation, A.O.; writing—review and editing, H.K. and M.Y.; visualization, A.O. and H.K.; supervision, H.K.; project administration, H.K. All authors have read and agreed to the published version of the manuscript.

**Funding:** Part of this work was financially supported by JSPS KAKENHI Grant Number: JP17K01442.

**Institutional Review Board Statement:** Not applicable.

**Informed Consent Statement:** Not applicable.

**Data Availability Statement:** Data sharing is not applicable to this article.

**Acknowledgments:** Kenneth Sandoval, Meredith Thornton, and Kathleen Duncan (The University of Oklahoma) provided information on the HIG FST4 1 strain and its characteristics.

**Conflicts of Interest:** The authors declare no conflict of interest.

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
