# Peer review of "Marine Biofilm Model Comprising a Loop-Type Biofilm Reactor and a Halomonas Strain HIG FST4 1, an Active Biofilm-Forming Bacterium"

_coatings, doi:10.3390/coatings12101605_

Round 1
Reviewer 1 Report
Authors developed and tested a loop-type laboratory biofilm reactor (LBR). The proposed system can be used to evaluate formation of the biofilm - Authors gave proof-of-concept using Halomonas strain.
The results can be published after minor revision:
- what is pure iron, pure aluminium used in the experiments? is is a chemically pure Fe and Al? why Authors didn't used steel or duraluminium sample?
- can wee see the picture of the experimental setup?
- Figure 2 - please add scale to the pictures.
- Figure 3: please move spectra to the top (as is done in Figure 4). It will be more readable.
- why Authors didn't performed any SEM or AFM observations of the surface of samples?
Author Response
Dear Dr. the reviewer,
We deeply appreciate spending your precious time and giving your kind and developing advices to our article. We revised the article according to your comments. We hope the revised article will be good quality to understand more easily and readable than original one.
We listed our answers to your comments as below.
- what is pure iron, pure aluminum used in the experiments? Is a chemically pure Fe and Al? Why authors didn't used steel or duraluminium sample?
Because we would like to avoid detecting “the side-effect” of other contents in steels and aluminum alloys on biofilm formation. In this study, we focused to understand the direct effect of Fe and Al on biofilm formation.
In previous our studies, we have researched the correlation with several metal elements to biofilm formation, and it is considered that ionized irons accelerate biofilm formation, while ionized aluminum does not affect biofilm formation because aluminum is not an essential element used in organisms (including bacteria). Additionally, aluminum is difficult to ionize because of easily passivated.
To explain the reason why we selected such three materials, the following sentence was added to revised manuscript.
(L.120–122)
Practically, steels and aluminum alloys are practically used for marine constructs and vessels, using these alloys had a risk that other compounds would affect biofilm formation. Therefore, we…….
- Can we see the picture of the experimental setup?
We added the photograph of this reactor as Figure 1a. Therefore, Figure 1 is consisted of (a): the photograph and (b): the illustration. Simultaneously, figure legend was also arranged.
(l.102–105)
Figure 1. Outline of the closed-circuit biofilm reactor. (a) and (b) indicate the photograph of practical reactor and the scheme of this reactor, respectively. In the illustration (b), arrows indicate the flow direction. Two necks of the three-necked beaker are connected to glass tubes, and the last (center position) is used for aseptic air exchange.
- Figure 2 - please add scale to the pictures.
Scale bars (black lines) were inserted into each microscopic image in Figure 2, and the explanation was added to the legend.
(L. 151–152)
Each black bar indicates 20 mm.
- Figure 3: please move spectra to the top (as is done in Figure 4). It will be more readable.
Thank you for this advice. We displayed original data about the Raman spectra of all samples (Al, Fe, and soda lime glass), and each “intensity” value showed original value. In the case of Fe-LBR and FE-TTC, the Raman data were almost overlayed under 1700 cm-1 region. Of course, we can technically move up the position of target spectra in Figure 3, however, we think overlaying data will be reflected to the similarity of biofilms between Fe-TTC and Fe-LBR. Therefore, Figure 3 remains original style.
- why Authors didn't perform any SEM or AFM observations of the surface of samples?
Because (1) we thought Raman spectroscopy was more suitable method to detect biofilms better than SEM, and (2) we did not have the observation technique of AFM.
SEM is traditionally used for identifying bacteria in biofilms (fortunately detect the EPS of biofilms). However, we consider “biofilms” are complex of bacteria, the EPS, and dead cell debris. During preparation of samples for SEM observation, biofilms will disappear and broken in structure, which often causes failure to observe biofilms. On the other hand, Raman spectroscopy can analyze biofilms without specific treatment such as completely dried and metal spattering, therefore we chose Raman spectroscopic analysis in this study.
Reviewer 2 Report
The subject of this study is biofilms which form in marine environments, which is a problem in indeed for vessels which navigate in the sea. In order to avoid dangerous bacteria which have been used frequently for related investigations in the laboratory, an alternative is presented in this study in order to reduce safety problems associated with the above mentioned bacteria. The Introduction describes the topic concise and in a way which is focused to the performed work. Three different substrates were employed in this work, which included materials from different chemical classes (ceramics and metals) and metals with different properties (iron and aluminum), thus providing a good selection for the investigation of the results (the reasons for the selection of these three materials are clearly and logically explained in the manuscript). The experiments, including the used materials, are well described in the Experimental Section and allow repetition by other groups. Biofilm formation was analyzed with optical microscopy (corresponding pictures of high quality are included in the manuscript) and Raman spectroscopy. The manuscript is fluently written and deserves publication after implementation of some modifications as outlined below.
p.4, Figure 2: Scale bars are missing in all optical microscope images.
p. 6 – 7, section 3.2.2.: This section should drastically shortened. In fact the noise between about 1200 cm-1 and 2500 cm-1 is too high to evaluate reliably peak positions, i.e. it is not really clear if there are peaks in this region at all in the blue and red spectrum (and in the black spectrum anyway, as already evident from text). Thus it should be written in the text that peak detection is highly uncertain in this spectral range due to comparably high noise, which indicates that the quantity of substances on soda lime glass is rather low (which is in agreement with the view of the authors). The corresponding “signal” attributions in Figure should therefore be deleted.
p. 8, Section 3.2.3.: Also this section should be somewhat shortened as Figure 5 shows that in the blue spectrum the “signals” between ca. 1400 cm-1 and 2100 cm-1 are in the range of the noise. Accordingly, the peak descriptions in the blue spectrum should be deleted in the range between 1400 cm-1 and 2100 cm-1.
p. 7, Figure 4, and p. 9, Figure 5: It is not explained in the caption to the figure which color (blue, red and black) is representative for which sample.
Author Response
Dear Dr. the reviewer,
We deeply appreciate spending your precious time and giving your kind and developing advices to our article. We revised the article according to your comments. We hope the revised article will be good quality to understand more easily and readable than original one.
We listed our answers to your comments as below.
- p.4, Figure 2: Scale bars are missing in all optical microscope images.
Scale bars (black lines) were inserted into each microscopic image in Figure 2, and the explanation was added to the legend.
(L. 151–152)
Each black bar indicates 20 mm.
- 6 – 7, section 3.2.2.: This section should drastically shortened. In fact the noise between about 1200 cm-1 and 2500 cm-1 is too high to evaluate reliably peak positions, i.e. it is not really clear if there are peaks in this region at all in the blue and red spectrum (and in the black spectrum anyway, as already evident from text). Thus it should be written in the text that peak detection is highly uncertain in this spectral range due to comparably high noise, which indicates that the quantity of substances on soda lime glass is rather low (which is in agreement with the view of the authors). The corresponding “signal” attributions in Figure should therefore be deleted.
Thank you for the expertise opinions. We deleted the peak descriptions during 1200–2500 cm-1 about blue line (after TTC test) and red line (after LBR test) from the text and Figure 4. Additionally, we added the reason why we except this range from peak recognition.
For Glass-TTC, the 1250–1288 cm-1, 1421–1494 cm-1, 1559–1883 cm-1, and 2033–2522 cm-1 peak were attributed to the C=O stretching vibration of peptide bonds in proteins [16], scissoring and twisting vibrations of CH2 in lipids [13], carbonyl compounds, a mix of C=C or C=O vibration in fatty acids, amide bond in proteins, and DNA strand bonds [13–15], respectively. The peaks at 2867–2936 cm-1 and 3044–3566 cm-1 denote fundamental vibrations in proteins and nucleic acids. For Glass-LBR, the peaks at 1279 cm-1, 1294–1428 cm-1, 1472–1507 cm-1, 1552–1888 cm-1, and 1932–2562 cm-1 were associated with amide III linkages in proteins [12], HC=CH in unsaturated fatty acids [13], CH2 group scissoring and twisting vibrations in lipids [13], carbonyl compounds (polysaccharides) [11], a mix of C=C or C=O vibrations in fatty acids, amide bonds in proteins, DNA strand bonds, and fundamental vibrations in proteins or nucleic acids [13–15], respectively. The fundamental vibrations on proteins or nucleic acids were denoted by the peaks at 2640–2723 cm-1, 2862 – 3007 cm-1, and 3243–3565 cm-1 [11].
(L.220–225)
Both of Glass-TTC and Glass-LBR, several small peaks at the range of 1200–2500 cm-1 are failed to peak detection due to comparably high noise, which indicates that the quantity of substances on soda lime glass is rather low. The fundamental vibrations on proteins or nucleic acids were denoted by two peaks at 2867–2936 cm-1 and 3044–3566 cm-1 for Glass-TTC, and by three peaks at 2640–2723 cm-1, 2862 – 3007 cm-1, and 3243–3565 cm-1 for Glass-LBR [11].
- 8, Section 3.2.3.: Also this section should be somewhat shortened as Figure 5 shows that in the blue spectrum the “signals” between ca. 1400 cm-1 and 2100 cm-1 are in the range of the noise. Accordingly, the peak descriptions in the blue spectrum should be deleted in the range between 1400 cm-1 and 2100 cm-1.
Thank you for the expertise opinions. We deleted the peak descriptions about blue line (after TTC test) in the range between 1400 cm -1 and 2100 cm-1 from the text and Figure 5. According to this preparation, we arranged the target sentences. However, the peak at 1434 cm-1 is remained because the sharpen peak is different from noise.
The peaks at 1288 cm-1 and 1434 cm-1 denoted the HC=CH bonds in unsaturated fatty acids. The peaks at 1672–1673 cm-1 and 1774–1880 cm-1 were attributed to the C=O stretch vibration in peptide linkages in proteins [16] and carbonyl compounds in polysaccharides [11], respectively. The peaks at 1949–2197 cm-1 and 2291–2407 cm-1 were associated with the mixture of C=C or C=O vibrations in fatty acids, amide bonds in proteins, and DNA strand bonds [13–15]. The peaks at 2851–2924 cm-1, 3207–3353 cm-1, and 3450–3580 cm-1 were attributed to the fundamental vibrations in proteins or nucleic acids [17].
(L.246–250)
The peaks at 1288 cm-1 and 1434 cm-1 denoted the HC=CH bonds in unsaturated fatty acids. The peak at 2291–2407 cm-1 was associated with the mixture of C=C or C=O vibrations in fatty acids, amide bonds in proteins, and DNA strand bonds [13–15]. The peaks at 2851–2924 cm-1, 3207–3353 cm-1, and 3450–3580 cm-1 were attributed to the fundamental vibrations in proteins or nucleic acids [17].
- 7, Figure 4, and p. 9, Figure 5: It is not explained in the caption to the figure which color (blue, red and black) is representative for which sample.
We added the following sentences in the legends.
(L.232–235, 264–267)
The black, red, and blue lines indicate the Raman spectra for iron piece prior to the tests, after LBR test, and after slowly rotating TTC test, respectively. P: proteins; n.a.: nucleic acids; h.a.: humic-like acids; L: lipids; S: polysaccharides.